# Whole genome analysis for 163 gRNAs in Cas9-edited mice reveals minimal off-target activity

Kevin A. Peterson [1,10], Sam Khalouei[2,7,10], Nour Hanafi [2], Joshua A. Wood[3,8], Denise G. Lanza [4], Lauri G. Lintott [5], Brandon J. Willis[3], John R. Seavitt[4], Robert E. Braun [1], Mary E. Dickinson [6], Jacqueline K. White[1], K. C. Kent Lloyd [3], Jason D. Heaney [4], Stephen A. Murray [1], Arun Ramani[2,9] & Lauryl M. J. Nutter [5✉]

Genome editing with CRISPR-associated (Cas) proteins holds exceptional promise for "correcting" variants causing genetic disease. To realize this promise, off-target genomic changes cannot occur during the editing process. Here, we use whole genome sequencing to compare the genomes of 50 Cas9-edited founder mice to 28 untreated control mice to assess the occurrence of *S. pyogenes* Cas9-induced off-target mutagenesis. Computational analysis of whole-genome sequencing data detects 26 unique sequence variants at 23 predicted off-target sites for 18/163 guides used. While computationally detected variants are identified in 30% (15/50) of Cas9 gene-edited founder animals, only 38% (10/26) of the variants in 8/15 founders validate by Sanger sequencing. In vitro assays for Cas9 off-target activity identify only two unpredicted off-target sites present in genome sequencing data. In total, only 4.9% (8/163) of guides tested have detectable off-target activity, a rate of 0.2 Cas9 off-target mutations per founder analyzed. In comparison, we observe ~1,100 unique variants in each mouse regardless of genome exposure to Cas9 indicating off-target variants comprise a small fraction of genetic heterogeneity in Cas9-edited mice. These findings will inform future design and use of Cas9-edited animal models as well as provide context for evaluating off-target potential in genetically diverse patient populations.

[1] The Jackson Laboratory, Bar Harbor, Maine, ME, USA. [2] The Centre for Computational Medicine, The Hospital for Sick Children, Toronto, ON, Canada. [3] Mouse Biology Program, University of California Davis, California, CA, USA. [4] Department of Molecular and Human Genetics, Baylor College of Medicine, Houston, TX, USA. [5] The Centre for Phenogenomics, The Hospital for Sick Children, Toronto, ON, Canada. [6] Department of Integrative Physiology, Baylor College of Medicine, Houston, TX, USA. [7] Present address: Personalis, Inc. 1330 O'Brien Drive, Menlo Park, CA, USA. [8] Present address: The Jackson Laboratory, Bar Harbor, Maine, ME, USA. [9] Present address: Invitae, San Francisco, CA, USA. [10] These authors contributed equally: Kevin A. Peterson, Sam Khalouei. ✉email: lauryl.nutter@sickkids.ca

CRISPR/Cas9 genome editing has tremendous therapeutic potential for treating human genetic diseases[1]. The widely used *S. pyogenes* CRISPR-associated protein 9 (Cas9) is a programmable RNA-guided endonuclease that can be targeted to precise locations in the genome of virtually any organism using a 20-nt protospacer sequence within a guide RNA (gRNA)[2]. The 20-nt target site must be immediately upstream of an NRG sequence referred to as the protospacer adjacent motif (PAM)[3] with an NGG site conferring increased cutting efficiency compared to NAG[4]. Given the potential number of matches and mismatches for a 20-nt sequence in large genomes, along with reports of off-target Cas9 mutagenesis in cultured cells[5], concerns regarding off-target Cas9 activity resulting in unintended genome modifications remain. In response, numerous methods have been developed to mitigate and detect purported off-target effects of Cas9 activity, such as the use of high fidelity Cas9 variants and gRNA modifications[6–10], and unbiased molecular approaches to assess Cas9 off-target cutting: BLESS[11], CIRCLE-seq[12], Digenome-seq[13], GUIDE-seq[14], and SITE-seq[15]. However, these detection methods are often difficult to implement in large-scale animal production scenarios and would be ethically unjustifiable in the absence of compelling arguments for their need. Further, the extent of reported Cas9-specific off-target mutagenesis varies across studies, ranging from nearly undetectable to moderate when reagents are delivered directly to mouse zygotes[16–18]. These studies typically involve whole-genome sequencing (WGS) for a limited number of gRNA targets with trios (parental-progeny) or intercrosses of inbred strains. While trio information enables discrimination of germline variation from Cas9 off-target events, it does not discriminate natural de novo variation in a colony of inbred mice and is not practical in mouse genetic engineering facilities where zygotes are pooled from multiple embryo donors. Thus, the identification of off-target mutations after animal model production is confounded by natural variation in genetically engineered mice. The Knockout Mouse Phenotyping Program (KOMP²) uses Cas9 for high-throughput mouse line production by genome editing to generate null alleles in the inbred C57BL/6N strain for broad-based in vivo phenotyping (mousephenotype.org). The KOMP² production pipeline thus provides a resource for evaluating off-target mutagenesis mediated by Cas9.

## Results

### Whole genome sequencing of Cas9-edited founder mice.
To assess the risk of off-target mutations when using Cas9 to create gene-edited mouse lines within the context of natural variation, we compared whole-genome sequencing for 28 untreated wild-type mice with 50 founder animals from the C57BL/6N isogenic background generated using protocols and guides designed to minimize off-target risk (Supplementary Data 1). Target genes were randomly selected from those in production in the KOMP² pipeline at the beginning of this work. All Cas9-derived knock-out mice were generated using a deletion approach that deleted a critical exon(s) using 2, 3 or 4 gRNAs per target gene. Collectively, the founders tested represented 163 different guide RNAs used across four KOMP² Centers (Fig. 1). To replicate typical procedures used in mouse genetic engineering facilities, control mice were randomly selected from the production center's wild-type C57BL/6N stud male colony used for embryo production or mice from embryos that were not exposed to Cas9 or gRNAs from the same embryo pool used to generate founders. WGS was performed on individual animals to an average depth of ~35-40X coverage and processed for variant calling (Supplementary Fig. 1).

### Sequence variant identification in Cas9-edited founder mice.
For analysis, we first applied a primary filter to remove any variants found in dbSNP or the European Variant Archive (EVA) and a secondary filter to eliminate any variants shared between any two mice (Fig. 1; Supplementary Data 2). This process identified a median of 1,115 unique variants per control mouse and 1,034 variants per Cas9-edited mouse (Fig. 2a). Of these, ~756 single nucleotide variants (SNVs) and ~276 insertion/deletion variants (indels) were found per control mouse and ~713 SNVs and ~322 indels per Cas9-edited mouse (Fig. 2a). No significant differences were observed for the total number or type of variants between control and Cas9-edited mice. Further, the genomic position of variants did not measurably differ between groups, with most variants found within intergenic regions and introns and a smaller number observed in exons (Fig. 2b). Most variants were heterozygous suggesting a large degree of diversity within mouse colonies (Fig. 2c). However, many homozygous variants ($n = 2876$) were shared between at least 10 different mice, highlighting a number of C57BL/6N-specific variants currently not found in dbSNP or EVA (Fig. 2d).

### Evaluation of Cas9-mediated off-target activity.
Our variant calling pipeline successfully confirmed the expected exon deletion in 49/50 founders (98%). The single missed deletion corresponding to *Rasgef1a* was successfully identified by LUMPY[19] and Manta[20], but was later filtered out by Manta due to low quality and thus failed to meet our threshold of being independently called by two or more programs (Supplementary Fig. 2).

Given the high concordance between WGS data and mutation detection, we set out to determine the extent to which unintended off-target variants were directly caused by spurious Cas9 activity. First, Cas-OFFinder[21] was used to identify all predicted off-target sites associated with NGG or NAG PAM sequences allowing up to 5 mismatches with one DNA and/or RNA bulge compared to the on-target cut site. This resulted in 555,032 potential off-target sites in the genome (mm10) for the 163 tested gRNAs (Supplementary Data 1). In our WGS data, we detected variant calls at 0.005% (26/555,032) of predicted off-target sites associated with 15/50 founders resulting in less than one variant at a predicted off-target site per founder animal (Supplementary Data 3 and Supplementary Figs. 3-11). However, we did identify several loci that had more than one variant overlapping a predicted off-target site in a founder animal. For example, three of four off-target variants identified in the *Tmem171* founder were SNVs and two of these three were within three bases of each other and associated with the same guide in a region appearing to be highly polymorphic (Supplementary Fig. 8b and Supplementary Data 3). One of the three variants was an indel while the other two were SNVs. Given the polymorphic nature of this genomic region, the SNVs may not be the result of Cas9 activity. In general, the majority (96%) of WGS-detected variants overlapping predicted off-target sites were in intergenic or intronic regions (Fig. 3a) and not in coding regions. These findings support the relative low risk of spurious Cas9 activity in these gene-editing experiments.

When we evaluated the relationship between the variant type, the PAM sequence and the number of mismatches, we found that NAG was primarily associated with structural variants and NGG with small indels (Fig. 3b). There was also an enrichment for indels, which are the most likely outcome of Cas9 activity, with 4 mismatches at sites with an NGG PAM. To determine if this trend was maintained at higher mismatch allowances, we increased the Cas-OFFinder mismatch allowance to 6 and repeated the analysis comparing variant type and associated PAM. While increasing the mismatch allowance did result in the identification of more indel variants in our WGS data, 76% (16/21) of indels were associated with off-target protospacer

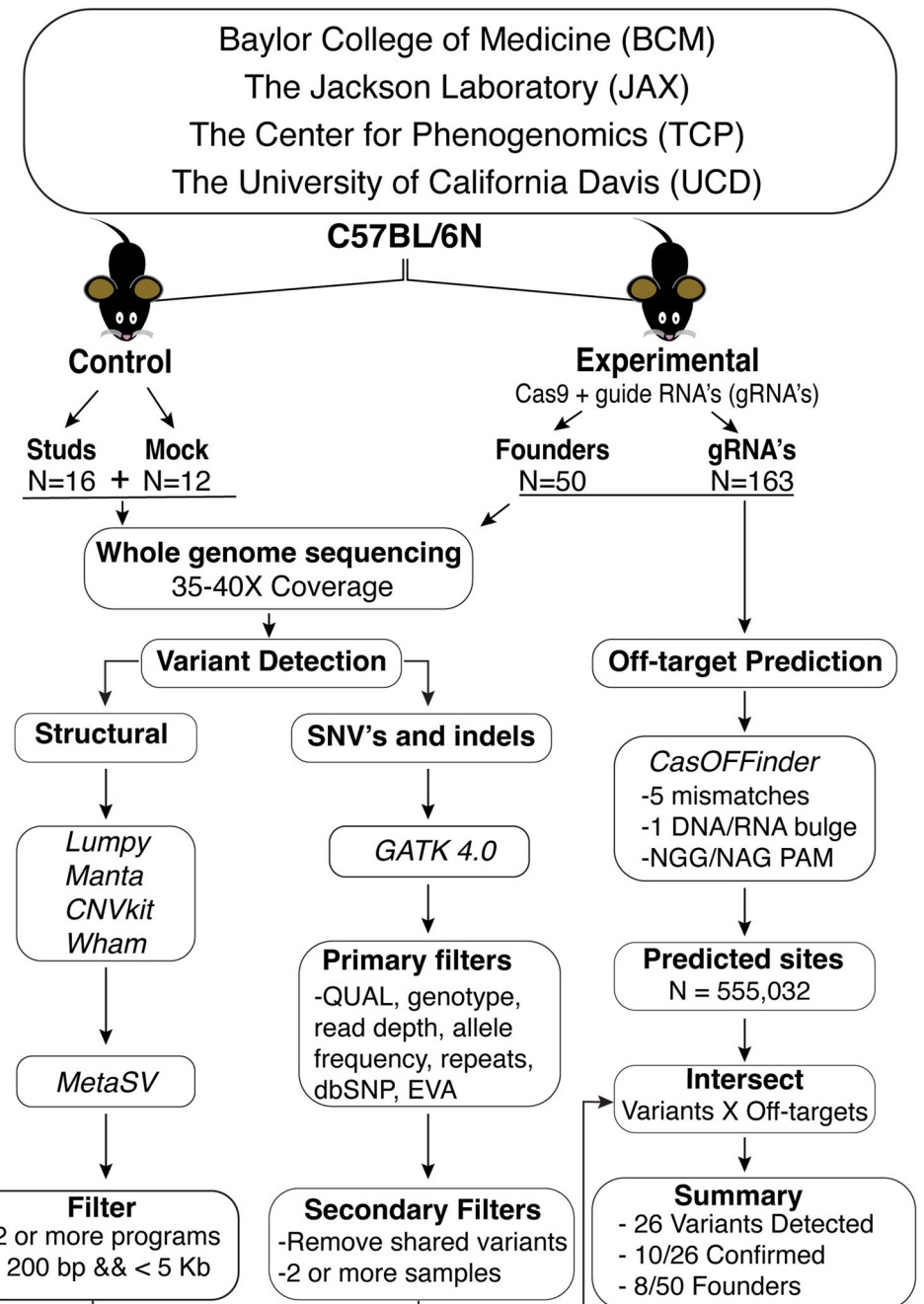

**Fig. 1 Multicenter analysis to assess off-target risk in Cas9-edited founder animals using whole genome sequencing (WGS).** Genomic DNA from a subset of C57BL/6N stud males used for embryo production or zygotes that were not treated with Cas9 was used as control DNA. Founders born from Cas9 editing experiments on zygotes from the control stud males or from the same embryo pool comprised the experimental group. Each founder animal was created using a multi-guide strategy to delete a critical exon(s). Founder animals were selected for WGS analysis after confirmation of germline transmission of the expected deletion. The whole-genome sequence analysis pipeline detected single nucleotide variants and small indels as well as potential structural variants. Potential off-target sites were predicted using Cas-OFFinder using permissive parameters and intersected with detected variants to identify putative off-target mutations.

sequences adjacent to NAG PAM sequences. There was also an increase in SNVs as well as structural variants at the 6-mismatch allowance (Fig. 3b).

To determine the likelihood that variants with four or more mismatches were due to Cas9 off-target activity, we compared the rate of variant detection in Cas9-edited mice to the rate of incidental overlap of variant calls in the control mice by randomly sampling from all Cas-OFFinder predictions using the median number of predicted Cas-OFFinder sites per sample. The median

number of sites was used in the analysis of control samples to account for differences in the total number of predicted off-target sites for a given experimental sample (set of gRNAs). This analysis identified 56 variants in the WGS data associated with a predicted off-target site in 25/28 of the control mice (Supplementary Data 4). Unlike the variants detected at predicted off-target sites in the Cas9-edited mice, all the predicted off-target sites in control mice had 5 or 6 mismatches and most were structural variants or SNV and not small indels as would usually

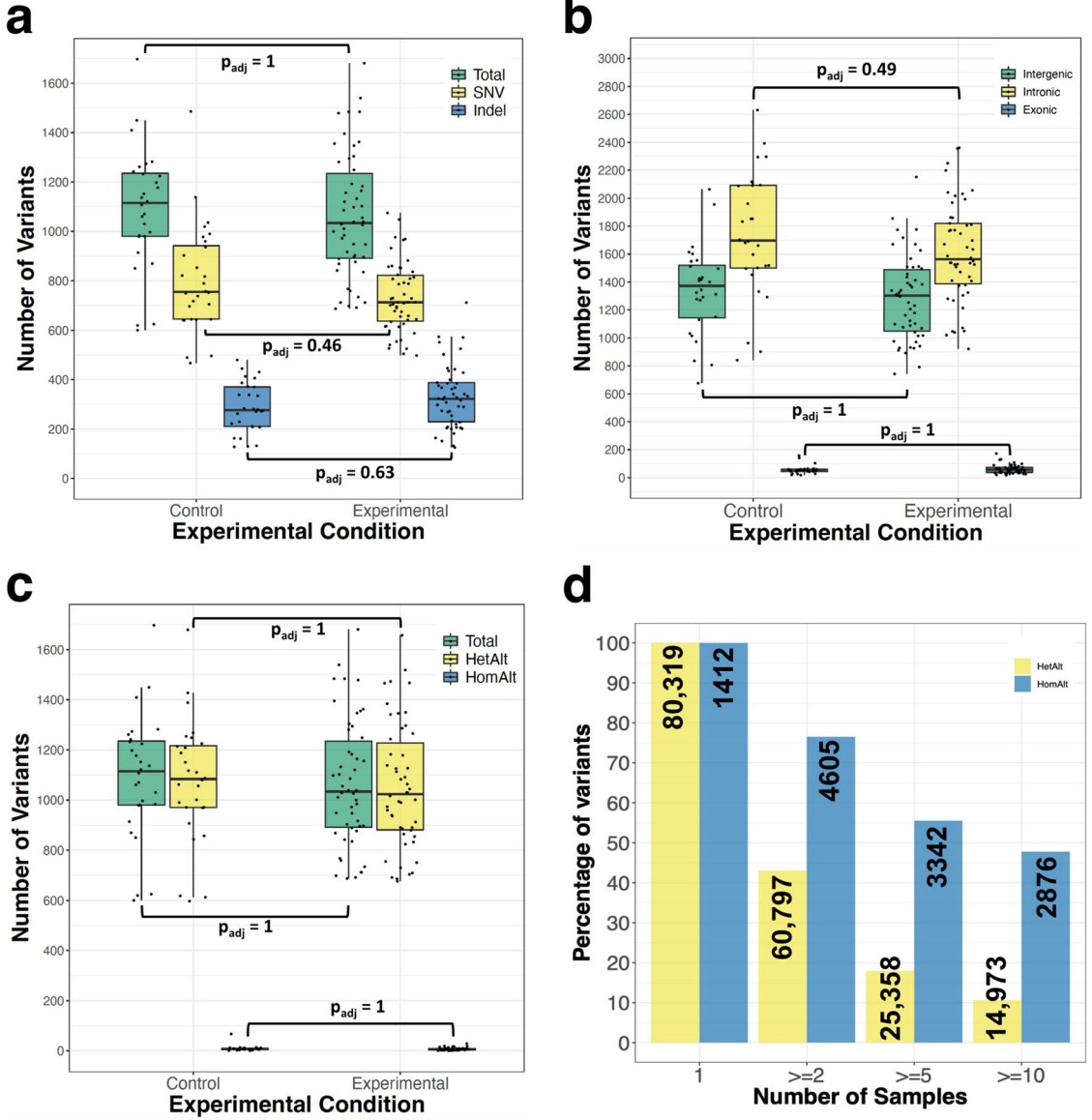

**Fig. 2 Summary of variants detected in *n* = 28 biologically independent control and *n* = 50 biologically independent Cas9-edited experimental mice.**
**a** Boxplots showing the total number of single nucleotide variants (SNV) and insertion-deletion (indel) variants identified within each experimental group.
**b** Distribution of variants throughout the genome relative to genic sequences. **c** Zygosity of SNV and indel variants identified. **d**. Percentage of variants
found in sample subsets. For boxplots: Lines within the boxes represent medians, lower edges the 1st quartile, and upper edges the 3rd quartile. The length
of the box corresponds to the interquartile range (IQR). Whiskers extend to the minimum or maximum values that fall within 1.5 X IQR of the 1st and 3rd
quartiles, respectively. Each datapoint represents the number of variants for a given sample. Statistical testing was done using Wilcoxon tests and
Bonferroni corrections, evaluated at α = 0.05. HetAlt, heterozygous variant; HomAlt, homozygous variant.

be expected from Cas9 activity (Supplementary Fig. 12a).
Furthermore, WGS-detected variants associated with predicted
off-target sites contained more variants at PAM distal positions
when allowing up to 5 mismatches which is consistent with the
reported mechanisms of Cas9 off-target activity[4]; however, this
trend was lost when increasing the allowance to 6 mismatches
(Supplementary Fig. 12b).

Given the potential for false positives at the higher mismatch
allowance, we used Sanger sequencing to validate indels
associated with off-target sites containing 5 or fewer mismatches
in non-polymorphic regions, and some structural variants where
the repetitive nature of the genomic sequence did not preclude
appropriate primer design (Fig. 3c,d and Supplementary Data 3).
Despite previous reports of structural variants resulting from
Cas9 activity[22], we were only able to confirm small indels at off-

target sites near an NGG PAM (Supplementary Data 3 and
Supplementary Figs. 3-11).

**In vitro assessment of Cas9 off-target activity**. While compu-
tational prediction of off-target sites can identify most undesired
Cas9 activity, there are examples where in-vitro methods detected
unpredicted off-target activity (e.g., Anderson et al.[16]). To
experimentally interrogate our Cas-OFFinder predictions,
CIRCLE-seq was used to test the in vitro off-target activity
associated with six different guides, three with (*Irf3*, *Lpgat1*, and
*Tmem171*) and three without (*Aimp1*, *Dusp15* and *Ptprk*) variants
detected at predicted off-target sites in our WGS analysis of
founder mice (Fig. 4a). CIRCLE-seq and Cas-OFFinder both
predicted seven off-target sites with evidence of editing in our

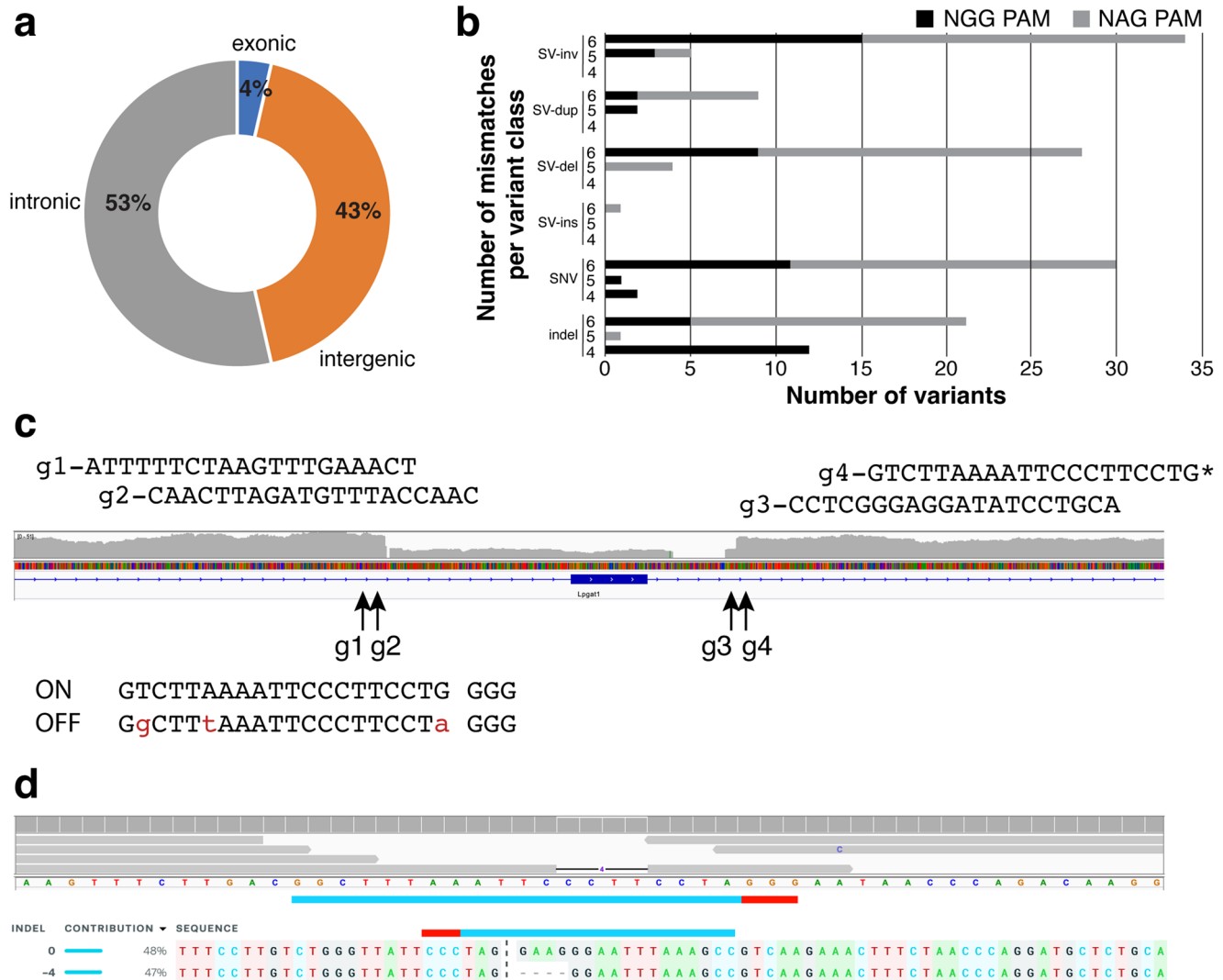

**Fig. 3 Detection of potential off-target Cas9 activity in whole genome sequencing data. a** Position of WGS-detected sequence variants associated with predicted off-target sites ($N = 26$) relative to genes with percentage for each shown in doughnut plot. **b** Classification of variants associated with WGS-detected predicted off-target sites, associated PAM sequence, and impact of increasing the number of allowed mismatches in Cas-OFFinder predictions from ≤4, 5 or 6. **c** On target identification of exon deletion at *Lpgat1* generated using a four-guide design strategy. Off-target Cas9 activity was associated with guide sequence, g4. Mismatch sites are shown in red lowercase letters. **d** Primary whole-genome sequence data used to identify off-target site and Sanger sequence validation from founder animal DNA confirming 4-bp deletion in founder. ICE analysis shown below predicts a heterozygous allele frequency (https://www.synthego.com/products/bioinformatics/crispr-analysis). Abbreviations: SV, structural variant; inv, inversion; dup, duplication; del, deletion; SNV, single nucleotide variant; indel, insertion/deletion.

WGS data. Cas-OFFinder identified 10 off-target sites with evidence of variants in our WGS data not identified by CIRCLE-seq while CIRCLE-seq uniquely identified two additional sites with WGS evidence of Cas9-editing, one each for *Dusp15* and *Tmem171* (Supplementary Fig. 11b,c). The remaining sites only predicted by CIRCLE-seq showed no evidence of Cas9-induced variation in our WGS data (Fig. 4b). These results support the conclusion that Cas9 off-target activity in gene-edited mice is rare and that the parameters used in this study for in silico prediction of off-target sites captured most of the potential in-vivo off-target activity. Collectively, these findings indicate that Cas9 off-target activity is predictable and can be minimized with careful guide selection.

**Analysis of genetic heterogeneity among C57BL/6N inbred mice.** To better understand the extent of genetic heterogeneity in inbred mice relative to the risk of Cas9 off-target activity, we analyzed the variants identified across all samples. Heat map analysis showed the largest factor contributing to the clustering of mice based on sequence variation was differences between the two C57BL/6N substrains used in this study (Fig. 5). Further, there was no difference in the number of variants (Fig. 2a-c) or clustering (Fig. 5) of Cas9-edited animals when compared to wild-type or untreated controls. Thus, when using appropriately selected guides, the diversity between any two individuals of the same substrain is greater than what may be introduced by potential Cas9 off-target activity.

**Discussion**
In this study, we set out to determine how frequently Cas9 off-target editing occurs in founder animals when applying well-defined design principles. While our WGS analysis pipeline identified variants in 30% (15/50) of our founders at Cas-OFFinder predicted off-target sites, only 10/26 (38%) off-target

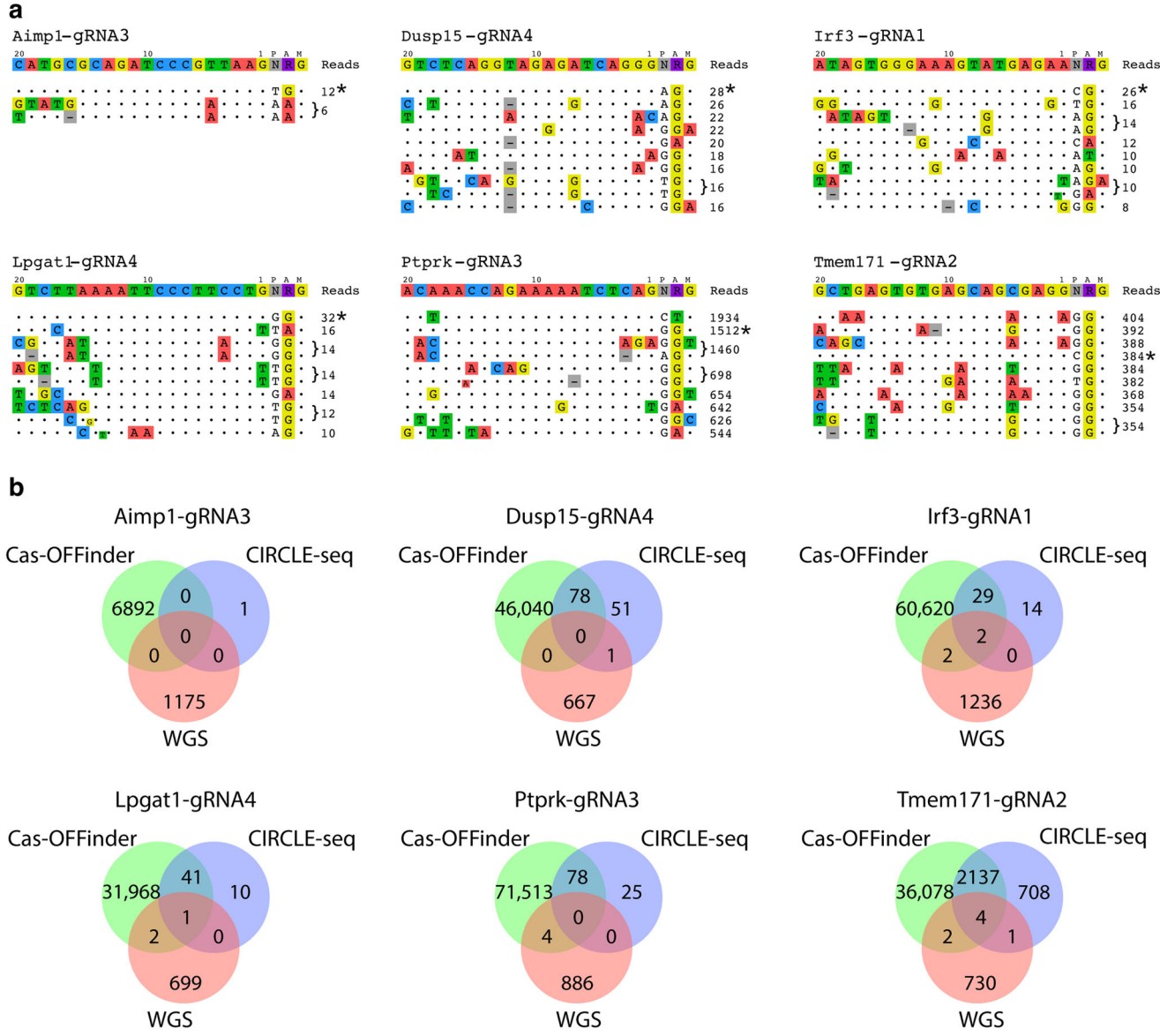

**Fig. 4 CIRCLE-seq analysis for select guides. a** Top 10 off-target sites for each guide identified using CIRCLE-seq. The on-target site is marked with an asterisk (*). **b** Venn diagrams for each guide tested showing overlap between Cas-OFFinder predicted off-targets, CIRCLE-seq and variants detected using whole-genome sequencing (WGS).

events, associated with 16% (8/50) of founders, could be confirmed by Sanger sequencing. Since validated off-target variants were not linked to the targeted genes, off-target mutations may be segregated via backcrossing that normally occurs during breeding for line expansion or during intercross of heterozygous mice to produce experimental and control cohorts. This breeding strategy controls for both naturally occurring and Cas9-mediated off-target variation. Furthermore, large-scale off-target structural variants were not confirmed, and the frequency of Cas9-induced off-target editing occurred far less often than naturally occurring sequence variation found between any two mice of the same substrain (<1 variant *cf.* >1,100 unique variants per mouse, respectively).

Here, we provided WGS data for a far greater number of gRNA target sites than in previous reports[16–18]. Our study design encompassed commonly employed methods, microinjection and electroporation, used to deliver Cas9 to the mouse zygote, and captured the intrinsic heterogeneity present in a given inbred strain as they are typically maintained at a vendor or an

accredited mouse breeding facility. A limitation of our study is that we could only analyze a single founder for each editing experiment precluding our ability to determine reproducibility of off-target Cas9 activity associated with a specific guide. Operationally, obtaining parental genomic information from the multiple breeding pairs needed to generate embryos for a gene-editing experiment is not feasible for typical production workflows. Extrapolating these findings to genetically heterogeneous patient populations demonstrates a need for patient-specific genomic and/or transcriptomic sequence data to enable accurate in silico prediction and/or in vitro determination of Cas9 or other genome editing tools' off-target activity.

Although inbred strains are assumed to be isogenic, the spontaneous, de novo mutation rate in mice is estimated to be ~25 SNVs and 1-2 indels per generation[23,24]. Therefore, even carefully maintained colonies will accumulate hundreds to thousands of spontaneous variants as they are expanded from foundation stocks. Our results show that the rate of Cas9-induced off-target editing in a carefully designed mouse production

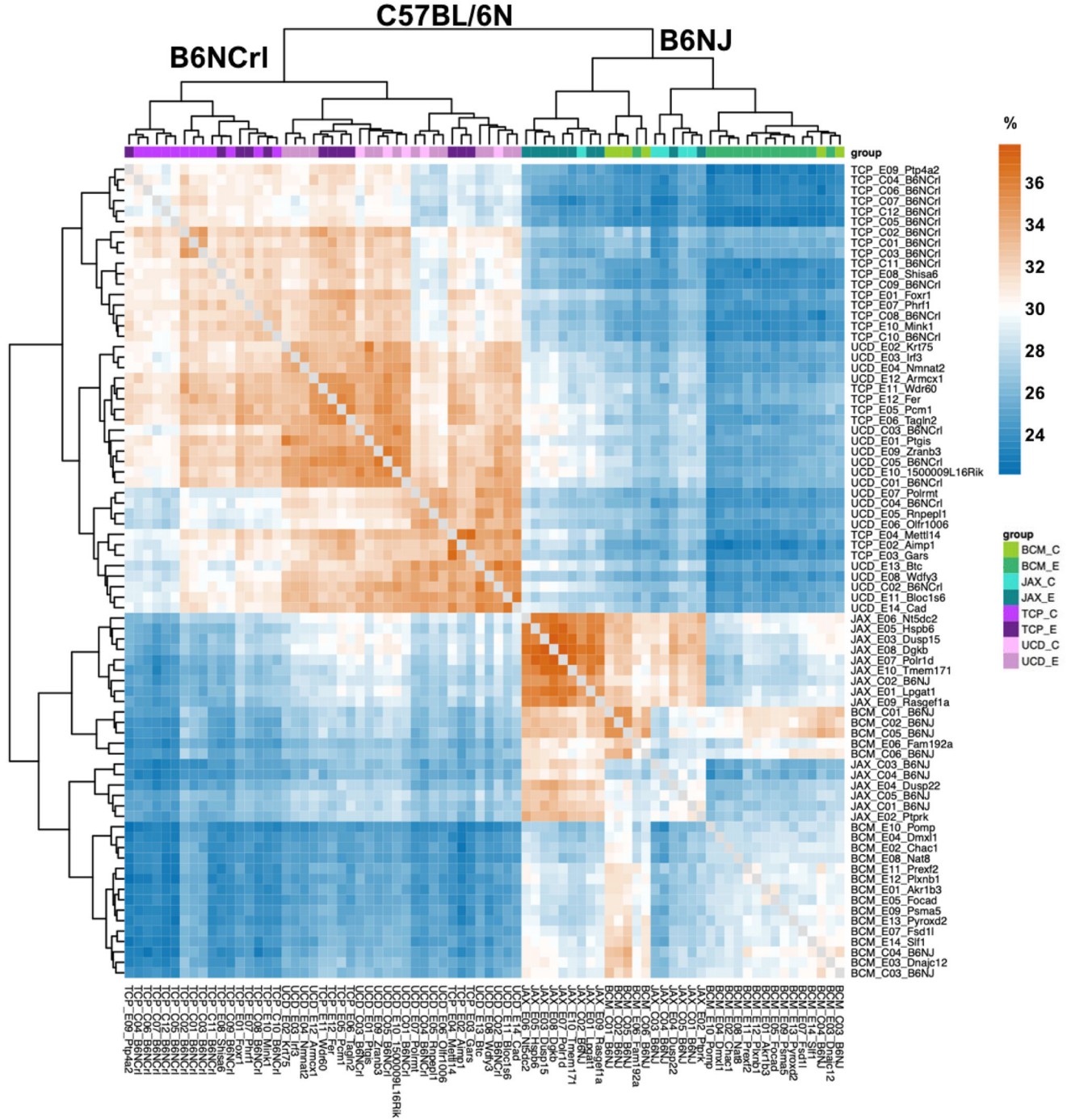

**Fig. 5 Genetic heterogeneity observed in individual mice of the same isogenic background.** Heatmap shows the percentage of common SNP variants and indels between mice highlighting two major clusters defined by animal production center and mouse substrain used for genetic modification. C57BL/6NCrl (NCrl) mice were used by The Centre for Phenogenomics (TCP) and University of California, Davis (UCD); while C57BL/6NJ (NJ) mice were used by The Jackson Laboratory (JAX) and Baylor College of Medicine (BCM). Sample names are shown on the bottom and right side of figure using the production center abbreviation followed by target gene or substrain background and designation of treatment group either control (C) untreated mice or experimental (E) Cas9-edited founders. Both treatment groups were interspersed with each other consistent with no statistical difference observed between control and experimental mice.

experiment is trivial relative to the overall genetic heterogeneity observed in inbred mouse colonies. Furthermore, the genetic bottlenecks that occur during genome engineering experiments will result in genetic drift from the original strain background. With this in mind, we strongly recommend the selection of appropriate genetic controls when assessing gene-phenotype relationships in genetically modified animals. As backcrossing

or outcrossing mice typically introduces more variation than Cas9, the appropriate control animals for most genetic experiments are littermate or line mate wild-type mice.

In summary, these data indicate that the risk of Cas9 cutting at predicted off-target sites is much lower than random genetic variation introduced into the genomes of inbred mice through mating. For gene editing experiments requiring the use of guides

that have increased off-target risk[16] or several off-target sites with fewer than three mismatches or no mismatches in the seed region[25], we advise checking for these events in both founder animals and in the N1 generation, particularly if the predicted off-target sites are genetically linked to the target or occur in an exon or functional sequence element.

## Methods

**Allele design and guide selection.** For multi-exon genes, a critical region (one or more exons) was identified as shared among all annotated full-length transcripts whose removal was predicted to result in a frame-shift mutation and introduction of premature stop codon greater than 50-nt from the final splice junction increasing the likelihood that the transcript(s) would be subjected to nonsense-mediated decay (NMD). gRNA sequences flanked the critical region and had no off-target sites with less than three mismatches adjacent to an NGG PAM. Guides were prioritized to minimize off-target risk and maximize predicted on-target cutting efficiency using prediction algorithms, including CRISPRtools[26], CRISPR MIT, CHOPCHOP[27], CRISPOR[28], and WGE[29]. Guide information is summarized in Supplementary Data 1.

**Animals.** All experiments were performed on C57BL/6N mice obtained from either The Jackson Laboratory (C57BL/6NJ; stock #5304) or Charles River (C57BL/6NCrl; strain code 027). All animals were maintained in accordance with institutional policies governing the ethical care and use of animals in research under approved protocols. All procedures involving animals at The Centre for Phenogenomics (TCP) were performed in compliance with the Animals for Research Act of Ontario and the Guidelines of the Canadian Council on Animal Care under Animal Use Protocols 0008, 0084 and 0275 reviewed and approved by TCP's Animal Care Committee. All animal use at Baylor College of Medicine, The Jackson Laboratory and UC Davis were done in accordance with the Animal Welfare Act and the AVMA Guidelines on Euthanasia, in compliance with the ILAR Guide for Care and Use of Laboratory Animals, and with prior approval from their respective institutional animal care and use committees (IACUC).

**Cas9 and guide RNA delivery to zygotes.** Gene editing was performed by either microinjection or electroporation of Cas9 mRNA and gRNA or Cas9 ribonucleoprotein complexes (RNP), respectively. Electroporation and microinjection experiments were conducted essentially as previously described[30–32]. Briefly, zygotes were collected from superovulated and mated C57BL/6N females into embryo collection media (Supplementary Data 6) and either manipulated immediately or incubated at 37 °C/5% $CO_2$ until manipulation and transfer. After manipulation, embryos were transferred into pseudopregnant recipients (Supplementary Data 6) identified by the presence of copulation plugs on the morning of transfer after mating overnight with vasectomized males.

**Founder identification.** Pups born after embryo transfer surgery were identified by ear tags (JAX, TCP), ear notches (JAX, BCM), or toe snips (UCD) and tissue biopsies were obtained according to approved institutional animal use protocols. DNA was isolated from tissue biopsies and subjected to end-point PCR to identify founders with the desired deletion (Supplementary Data 6).

**Germline transmission test breeding.** Founders were backcrossed to mice of the same substrain (Supplementary Data 6) at 6-8 weeks of age. Pups from this first backcross (N1) were genotyped using the same PCR conditions as founders and Sanger sequencing of the deletion amplicon was used to determine the definitive allele sequence. After sufficient N1 mice were produced to establish the line, founders were humanely euthanized according to approved animal use protocols and tissues collected for DNA extraction.

**Whole genome sequencing.** Genomic DNA for founder and control samples was extracted from spleens, tail tips or ear punches using phenol:chloroform or kit according to manufacturer's suggestions. DNA was quantified using fluorescence-based detection on a Qubit (Thermofisher) or by UV absorbance. Whole genome sequencing libraries were prepared at The Centre for Applied Genomics (The Hospital of Sick Children, Toronto, Ontario) following standard practices. Briefly, 700 ng of genomic DNA was sheared to an average size of 400 bp using a Covaris LE220 and was used as input to generate a whole-genome library using the TruSeq PCR-free kit (Illumina). The resulting DNA libraries were sequenced on Illumina Hi-seq X instrument to generate 2x150bp paired-end reads.

**NGS data analysis.** Sequence read quality was assessed using FastQC (http://www.bioinformatics.babraham.ac.uk/projects/fastqc/) and fastQ Screen[33], reads were processed through the bcbio pipeline (https://github.com/bcbio/bcbio-nextgen, ver. 1.1.0) for all steps from alignment to variant calling. Briefly, reads were aligned to mouse genome assembly (GRC Build 38/mm10) using bwa-mem[34] resulting in ~35-40X genome coverage for each sample. GATK 4.0 (Genome Analysis

Toolkit)[35,36] was used to call variants with the default parameters of the pipeline. The resulting variant call format (VCF) files were filtered to retain variants with QUAL > 30, DP > 9, GQ > 30 and AF > 0.1 using bcftools (ver. 1.6)[37]. Repetitive intervals were padded by two base pairs on either side to improve filtering due to indel variants that overlap boundaries of the repetitive intervals, and variants in repetitive and tandem-repeat regions were filtered out. Subsequently, variants were filtered using the central repository for mouse, European Variation Archive (EVA, https://www.ebi.ac.uk/eva), and dbSNP. A non-redundant set of variants was obtained by merging EVA files (GRCm38.p2, GRCm38p3, and GRCm38p4) that were then applied to filter out any common variants present in the 78 VCF files. Using a custom python script, heterozygous variants with a ratio of alternative alleles to total number of alleles less than 0.2 were excluded. Callable intervals were defined by bcbio pipeline for each sample based on the corresponding bam file. Since the number of variants for each sample, which is the main parameter in our analysis, can be influenced by the extent of callable intervals, we set to limit the primary-filtered VCF files to the intersection of all samples callable intervals to avoid any potential bias. Callable interval filtering was performed using a combination of custom-made scripts and bedtools *multiinter* tool (ver. 2.27.1)[38]. The intersection of callable intervals common to all 78 samples was used to filter the variants outside these intervals. A final set of variants was determined by applying a secondary filter to eliminate any variants observed in two or more independent samples using the bcbio-variation-recall ensemble software (https://github.com/bcbio/bcbio.variation.recall, ver. 0.1.7). bcftools *isec* was then applied to filter out the ensemble file variants from each sample's VCF file to create the secondary-filtered VCF files. Structural variants (SV's) were called by lumpy[19], manta[20], CNVkit[39], and Wham[40] followed by Metasv[41]. All samples successfully passed the MetaSV step, except for the *Nat8* sample, which encountered an unresolved technical error, and consequently was omitted from the rest of the SV analysis. After removing./. and 0/0 genotypes, SVs considered for final analysis had to be called by 2 or more methods and have at least 3 reads supporting with a variant length greater than 200 bp but less than 5 kb. SV secondary filtering to omit shared variants was performed using bedtools *pairtopair* (v2.26.0), using the non-default arguments '–type neither', and '–is' to ignore strandedness. SnpEff (ver. 4.3t)[42] was used to divide each sample's secondary filtered variants into intergenic, exonic, and intronic regions. The downstream and upstream variants were included in the intergenic category. The bcftools *isec* tool was used to find the number of overlapping and unique variants between any given two primary-filtered VCF files. The primary-filtered VCF files (excluding the dbSNP150, EVA, and callableIntervals filters mentioned above) were submitted to the European Variation Archive (EVA) database (https://www.ebi.ac.uk/eva/?eva-study=PRJEB61387). Predicted off-targets were identified by intersecting Cas-OFFinder[21] (max mismatch ≤ 6, DNA bulge ≤1, RNA bulge ≤1) sites with SNP/indel and SV VCFs for each sample, using bedtools *intersect* and then visually assessing identified in the VCFs using IGV (Integrated Genome Viewer). Predicted off-target sites that located in the mitochondrial chromosome or non-canonical contigs were excluded. Heatmap dendrogram was created by providing the percentage of common variants between any two samples (using "bcftools isec" tool and python scripts) as input to the pheatmap R package (https://cran.r-project.org/web/packages/pheatmap/).

**Off-target validation.** PCR products were generated from founder and control DNA samples using specific primers surrounding the region of interest and PCR conditions specified by enzyme suppliers for the applicable amplicon sizes (Supplementary Data 3 and Supplementary Data 6). PCR products were purified and submitted for Sanger Sequencing. The resulting control and edited .ab1 files and the guide sequence were used as input to ICE (https://www.synthego.com/products/bioinformatics/crispr-analysis). Screens shots for results shown indicate the analysis succeeded.

**CIRCLE-seq.** Experimental identification of off-target sites using CIRCLE-seq was performed as previously described[43]. Briefly, C57BL/6NJ genomic DNA was isolated from spleen and sonicated to average size of 300 bp using Covaris E220. The sonicated DNA was cleaned up with SPRI beads (Beckman Coulter) and prepared for circularization by performing end repair, A-tail and stem-loop adaptor ligation (Roche). Adaptor ligated DNA was digested with lambda exonuclease and exonuclease I followed by USER digest and treatment with PNK (NEB). Phosphorylated DNA was ligated overnight and then digested with Plasmid-safe ATP-dependent DNase (Lucigen) to eliminate non-circularized molecules. Circularized DNA was incubated with sgRNA and Cas9 (NEB) RNP complexes. Sequencing libraries were generated from linearized DNA and paired-end sequenced (2X150bp) on miSeq (Illumina). Data was processed using *circleseq* script (https://github.com/tsailabSJ/circleseq).

**Statistics and reproducibility.** No pre-calculation of sample sizes was done. This study analyzes whole genome sequence data from founder mice generated using 2 to 4 gRNAs per mouse for a total of 163 guides in 50 founder mice from 4 different centres. Between 5 and 12 control mice were analyzed at each centre (total $n = 28$). To our knowledge, this is the largest gRNA sample size evaluated for in vivo off-target Cas9 activity. Variation between control and experimental data sets was similar, so we concluded that the number of controls was sufficient for the size of

the experimental data set. No biological or technical replicates were used in this study. Data sets from all centres provided similar results supporting the conclusions of the study.

Embryos from a pool of collected embryos were randomly allocated to groups for treatment with different Cas9-gRNA combinations on each experimental day. When done, unmanipulated embryos or those electroporated without Cas9 were randomly assigned from the same pool of embryos as the manipulated embryos. Cas9-treated founders were selected based on whether the targeted deletion was detectable by end-point PCR across the target deletion region. This screen confirmed the activity of Cas9 within the embryo of the founder selected for analysis. With the exception of *Pomp*, all founders transmitted the target deletion allele through their germline and mouse lines with the deletion alleles were established. Whole genome sequencing was provided as a service at The Centre for Applied Genomics. Technicians there were blinded to sample type. Blinding at analysis was not possible as the groups needed to be assessed based on how the samples were or were not treated before sequencing.

To compare medians between experimental ($n = 50$) and control ($n = 28$) groups, the "compare_means" method of R package ggpubr (https://rpkgs.datanovia.com/ggpubr/) was used to perform Wilcoxon tests followed by Bonferroni correction. Boxplots where generated using ggplot2's (https://ggplot2.tidyverse.org/) "geom_boxplot" function. Datapoints were overlaid on boxplots using "position_jitterdodge", with the random seed set to 1 to ensure reproducibility of the boxplots. Random sampling from the pool of Cas-OFFinder predictions was performed using the R package dplyr (https://dplyr.tidyverse.org/), "sample_n" command. A random seed from 1 to 28 was set prior to each sampling, to create a reproducible randomly sampled collection of Cas-OFFinder predictions for each of the 28 control samples. No data was excluded from the study. However, due to a technical issue, one sample, *Nat8*, was omitted from complete SV analysis due to an unresolved technical error.

**Reporting summary**. Further information on research design is available in the Nature Portfolio Reporting Summary linked to this article.

## Data availability

Whole genome and CIRCLE-seq sequence data associated with this study were deposited to the NCBI Sequence Read Archive (SRA) under accession number PRJNA687003. The primary-filtered VCF files (excluding the dbSNP150, EVA, and callableIntervals filters mentioned above) were submitted to the European Variation Archive (EVA) database (https://www.ebi.ac.uk/eva/?eva-study=PRJEB61387). The source data Figs. 2 and 3b are provided in Supplementary Data 5. All other data supporting the findings are either presented as Supplementary Data or may be obtained from the authors (LMJN) on reasonable request.

## Code availability

Custom data parsing and filtering scripts are available at https://github.com/The-Centre-for-Phenogenomics/Cas9-WGS and Zenodo at https://doi.org/10.5281/zenodo.7823655.

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

## Acknowledgements

We would like to thank The Centre for Applied Genomics for generating whole genome sequencing data. We also thank the model production staff at The Jackson Laboratory, The Centre for Phenogenomics, Baylor College of Medicine, and UC Davis Mouse Biology Program for their assistance generating the founder mice used for this study. Research reported in this work was supported by the NIH Common Fund, National Human Genome Research Institute UM1 HG006348 (JDH and MED), and National Institutes of Health Office of the Director UM1 OD023221 (KCKL) and UM1 OD023222 (REB, SAM and JKW) and Genome Canada and Ontario Genomics OGI-137 (LMJN). Additional support was provided by the Canadian Center for Computational Genomics (C3G), part of the Genome Technology Platform (GTP), funded by Genome Canada through Genome Québec and Ontario Genomics (AR). The content is solely the responsibility of the authors and does not necessarily represent the official views of the National Institutes of Health.

## Author contributions

J.A.W., J.R.S., J.D.H., S.A.M., A.R., and L.M.J.N. conceived the project; J.D.H., D.G.L., B.W., J.A.W., K.A.P., K.C.K.L., S.A.M. and L.M.J.N. produced Cas9 knockout mice; A.R., K.A.P., N.H. and S.K. analyzed the data. D.G.L., L.G.L., B.W., K.A.P., and L.M.J.N. performed off-target validation experiments. K.A.P., S.K. and N.H. generated figures and drafted the manuscript. J.R.S., R.E.B., M.E.D., J.K.W., K.C.K.L., J.D.H., S.A.M., and L.M.J.N. received funding and supervised the research. All authors provided feedback and edits to the manuscript.

## Competing interests

The authors declare no competing interests.

## Ethics and Inclusion Statement

As described in the Animals section, all work with animals underwent ethical review by the affiliated institutional animal care and use committee (IACUC) or animal care committee (ACC). The authors agreed upon their respective roles in the research before the research began or at the time a new collaborating author joined the research team, and collectively represent all the institutions at which the research was conducted. This research did not require any exemptions to be completed, did not include human research participants, nor did it involve health, safety, security, or other risks to the researchers. All mouse lines generated during the research are available and can be accessed through the MMRRC (https://www.mmrrc.org/) or CMMR (http://www.cmmr.ca/). Citations relevant to the research were identified with PubMed searches using keywords relevant to the research undertaken.
