## [Peer Review File · Communications Biology]

Reviewers' comments:

Reviewer #1 (Remarks to the Author):

Peterson and colleagues investigated off-target activities of 163 guide RNAs (gRNAs) in Cas9-edited mice combining whole-genome sequencing (WGS) and computational prediction. The Cas9-induced off-target activity is one of the significant concerns that retard the CRISPR application, especially in clinics. Although many off-target identification assays and computational models have been developed, WGS is still the golden standard to comprehensively investigate the off-targets in the unknown region across the genome if the sequencing depth is high enough. The author used WGS to examine 50 Cas9-edited founder mice in this study and compared them with the corresponding control animals. They concluded that the Cas9-induced mutagenesis is rare compared to the natural genetic variations. The study provided helpful information to understand the potential risk better when using CRISPR engineered mice as animal models.

Major issues:

- 1, From previous off-target analysis, the "hot-spot" off-targets typically accounted small fraction off-targets, while most off-targets happened in very low frequencies, which brought concern about the sensitivity of the off-target identification. Also, this experiment used 50 mice to examine 163 gRNAs, which brought up another concern on the reproducibility of the results. It would be meaningful to examine a couple of gRNAs from multiple biological replicate animals (PMID: 29786090).
- 2, Another efficient approach to generate an off-target candidate list would be performing enrichment-based off-target identification assays (e.g., GUIDE-seq, DIGENOME-seq), which could verify the off-targets identified from the WGS.
- 3, The Cas-OFFinder parameter is set as "up to 5 mismatches". However, both GUIDE-seq and DISCOVER-seq have identified off-target with 6~7 mismatches.
- 4, The study included two editing materials, one the Cas9 mRNA and gRNA, the other is RNP. Also, two delivery methods, microinjection and electroporation, were used. And all these factors may largely affect off-target activities. The author should consider and reason why these data could be put together for analysis.

Minor issues:

- 1, Fig4, the authors concluded that "the largest contributing factor was the differences between the two C57Bl/6N substrains used in this study", it will be better to clarify what kind of difference of substrain is provided.
- 2, The method of identification of off-targets with Cas-OFFinder is obscure and more details should be provided. What is the input of the software, WGS data or just the genome reference?

Reviewer #2 (Remarks to the Author):

BRIEF SUMMARY

This paper traces a familiar path:

- mutagenise mouse zygotes with gRNAs
- WGS mutants
- variant call and filter (using controls)
- infer potential crispr off-target mutations by checking variant location vs expected crispr OT positions
- declare off-target mutations in their designs are either very rare or absent

This is the new data / design the authors bring:

- larger range of gRNAs (tens of target genes, instead of one or two)
- larger number of KO animals (50 compared to 10, say) and controls (28 compared to 10)
- exon ablation design instead of small indels

Here is where the authors compromise in comparison to other WGS approaches:

- The ability to know whether a mutation was present in a (wild-type) parents of the founder. The authors state this is not practical at their throughput scales, and that is reasonable assertion.

Here is where the authors compromise in comparison to in vitro approaches (circleseq etc):

- they have to infer an offtarget mutation instead of making a direct measurement.

OVERALL IMPRESSION

This work is sound, subject to the comments below.

SPECIFIC COMMENTS

1 The authors state "These studies typically involve whole-genome sequencing for a limited number of gRNA targets with trios (parental-progeny) or intercrosses of inbred strains; and, thus diminish our ability to generalize these findings or interpret off-target events in the context of natural variation". I agree that a limited number of gRNAs makes it harder to generalise findings. I disagree that it makes it harder to interpret OT events given natural variation. In fact, the trio designs the authors refer to make it much easier to discount natural variation, in comparison to author's design. This assertion should be rephrased to reflect this.

2 The authors really should state that their design is confounded by the expected natural generation-to-generation variation of the founders compared to the controls. Their approach discounts this variation (because it focuses on expected OT locations) but it's still there.

3 The authors should show a table of variant counts by mouse at each step of filtering, for instance the columns could be

unfiltered variants from GATK

filtered variants after QUAL , DP etc filters

filtered variants after repeat proximity

filtered variants after EVA etc removal

filtered variants after restrictions to callable regions

My guess is that they had a few hundred thousand WGS variants per mouse to start with, so they should be clear what the effects of each step were, that reduced the tally to about 1000 per mouse.

4 The authors state these two filters: $QUAL > 30$, $DP > 9$, $GQ > 30$ and $AF > 0.1$ and later "heterozygous variants with a ratio of alternative alleles to total number of alleles less than 0.2 were excluded". Is the "AF" actually VAF, and is that the same quantity as the ratio stated later? Please clarify.

5 Are the authors confident that the default parameters in their GATK pipeline will detect mosaic alleles (eg 2 cell stage editing) that form the many of the crispr-induced variants in a founder mouse?

6 The authors state in some detail the characteristics of the (putative crispr-induced) variants found in 28 of the 556k predicted off-target sites. There are ~1000 filtered variants in every founder and control mouse. Could the authors please show that the 28 variants they found in 556k predicted OT sites were unlikely to have been situated there by chance? In particular, they only discuss variants in the 50 KO founders. Does that mean that NO variants were found at off-target sites in any of the 28 control animals? If so, that would be compelling evidence that the variants were crispr induced. Please clarify.

Reviewer #3 (Remarks to the Author):

The authors conducted whole genome sequencing to evaluate off-targets in 50 founder mice that were subjected to SpCas9 genome editing. This is important as more data is still required to understand the scope of off-targets in Cas9-edited Zygotes.

The authors applied the secondary filter: "a secondary filter to eliminate any variants shared between any two independent samples". As Cas9 off-targets are largely sequence-dependent and recurrent, meaning that off-targets are expected to present in more than one individual, what were the rationales and justifications for this secondary filter?

"While we did computationally detect unintentional Cas9-mediated off-target activity in 36% (18/50) of our lines, only 9/28 (32%) off-target events, associated with 16% (8/50) lines, were confirmed." Please clarify what "computationally detect" mean exactly, and what was the confirmation method.

There is a typo "CRISRPR".

22 February 2023

RE: Manuscript COMMSBIO-21-2206-T

Dear Reviewers,

Thank you for taking the time to review our manuscript and provide your critical reviews. Below we provide a point-by-point response to your thoughtful feedback. Where appropriate, we have provided the page numbers in the manuscript file where we have changed or added text or figures to address your comments.

Reviewer #1:

Major issues:

1, From previous off-target analysis, the "hot-spot" off-targets typically accounted small fraction off-targets, while most off-targets happened in very low frequencies, which brought concern about the sensitivity of the off-target identification. Also, this experiment used 50 mice to examine 163 gRNAs, which brought up another concern on the reproducibility of the results. It would be meaningful to examine a couple of gRNAs from multiple biological replicate animals (PMID: 29786090).

RESPONSE: We appreciate this concern and agree that performing replicate experiments could be informative but is not practical in a large-scale production scenario. As this study has been ongoing for several years, we do not have access to replicate animals and it would not be ethical from an animal use perspective to generate additional samples to address this concern. We have made a comment to this in the Discussion for future studies and as limitation of the current study.

2, Another efficient approach to generate an off-target candidate list would be performing enrichment-based off-target identification assays (e.g., GUIDE-seq, DIGENOME-seq), which could verify the off-targets identified from the WGS.

RESPONSE: We applied CIRCLE-seq to experimentally determine the off-target activity for 6 of the 163 guides used in this study. We selected CIRCLE-seq over other approaches as they are not feasible in our cell-type of interest (zygote).

3, The Cas-OFFinder parameter is set as "up to 5 mismatches". However, both GUIDE-seq and DISCOVER-seq have identified off-target with 6~7 mismatches.

RESPONSE: We thank the reviewer for raising this point and have updated our off-target prediction using CasOFFinder to allow for 6 mismatches. We observed that increasing the number of mismatches to 6 resulted in a change in the profile of off-target protospacer sequences as well as the type of variants found at these locations. We have included the results and a discussion of them in the revised manuscript (pg. 6).

4, The study included two editing materials, one the Cas9 mRNA and gRNA, the other is RNP. Also, two delivery methods, microinjection and electroporation, were used. And all these factors may largely affect off-target activities. The author should consider and reason why these data could be put together for analysis.

RESPONSE: We recognize that different approaches were used to introduce Cas9 to the zygote but consider this a strength of the study rather than a limitation. These complementary datasets provide a more accurate

representation of the different methods currently in use in animal production centers. While we have seen a convergence in methods by groups affiliated with the Knockout Mouse Phenotyping Program (KOMP2), we think this data is of value to the broader research community. We have added a comment to the manuscript regarding this concern (pg. 8).

Minor issues:

1, Fig4, the authors concluded that "the largest contributing factor was the differences between the two C57Bl/6N substrains used in this study", it will be better to clarify what kind of difference of substrain is provided.

RESPONSE: We have added further clarification to address the nature of these differences.

2, The method of identification of off-targets with Cas-OFFinder is obscure and more details should be provided. What is the input of the software, WGS data or just the genome reference?

RESPONSE: We have updated the methods to provide additional details.

Reviewer #2:

1) The authors state "These studies typically involve whole-genome sequencing for a limited number of gRNA targets with trios (parental-progeny) or intercrosses of inbred strains; and, thus diminish our ability to generalize these findings or interpret off-target events in the context of natural variation".

I agree that a limited number of gRNAs makes it harder to generalise findings. I disagree that it makes it harder to interpret OT events given natural variation. In fact, the trio designs the authors refer to make it much easier to discount natural variation, in comparison to author's design. This assertion should be rephrased to reflect this.

RESPONSE: We agree with the reviewer and have modified the text to account for this point (pg. 3).

2) The authors really should state that their design is confounded by the expected natural generation-to-generation variation of the founders compared to the controls. Their approach discounts this variation (because it focuses on expected OT locations) but it's still there.

RESPONSE: We thank the reviewer for raising this point and have added a comment to address this concern (pg. 4).

3) The authors should show a table of variant counts by mouse at each step of filtering, for instance the columns could be

unfiltered variants from GATK

filtered variants after QUAL, DP etc filters

filtered variants after repeat proximity

filtered variants after EVA etc removal

filtered variants after restrictions to callable regions

My guess is that they had a few hundred thousand WGS variants per mouse to start with, so they should be clear what the effects of each step were, that reduced the tally to about 1000 per mouse.

RESPONSE: We have included a new supplemental table (Table S2) to account for the filtering steps.

4) The authors state these two filters: $QUAL > 30$, $DP > 9$, $GQ > 30$ and $AF > 0.1$ and later "heterozygous variants with a ratio of alternative alleles to total number of alleles less than 0.2 were excluded". Is the "AF" actually VAF, and is that the same quantity as the ratio stated later? Please clarify.

RESPONSE: We apologize for the confusion and include further clarification. AF and VAF are not the same. AF (allele frequency) refers to the field in the VCF output by GATK, and is calculated by dividing the allele count (AC) by allele number (AN). VAF is the variant allele fraction and represents the number of reads supporting the alternate variant at a particular site divided by the total number of reads supporting that site (ref + alt).

5) Are the authors confident that the default parameters in their GATK pipeline will detect mosaic alleles (eg 2 cell stage editing) that form the many of the crispr-induced variants in a founder mouse?

RESPONSE: Mosaicism is a potential confound in any study analyzing a founder population of animals edited by Cas9 and we have done our best to account for this. Determining the biological significance of where this cut-off should be set is not one that is well established in the field. The cut-off in this study (0.2) should detect edits introduced at the 2-cell stage where four copies of the genome are present and an edit at any one site would result in a 25% allele frequency.

6) The authors state in some detail the characteristics of the (putative crispr-induced) variants found in 28 of the 556k predicted off-target sites. There are ~1000 filtered variants in every founder and control mouse. Could the authors please show that the 28 variants they found in 556k predicted OT sites were unlikely to have been situated there by chance? In particular, they only discuss variants in the 50 KO founders. Does that mean that NO variants were found at off-target sites in any of the 28 control animals? If so, that would be compelling evidence that the variants were crispr induced. Please clarify.

RESPONSE: We have re-analyzed the control samples for overlap between variants and predicted off-target sites (now allowing up to 6 mismatches) to estimate our potential False Discovery Rate. In our 28 samples, we did detect variants associated with predicted Cas9 off-target site; however, all of these were associated with sites of 5 or 6 mismatches and were either structural variants or SNV not indels. We discuss these findings in the revised manuscript (pg. 6) and the results of this analysis are now included in Supplemental Table S5.

Reviewer #3:

The authors applied the secondary filter: "a secondary filter to eliminate any variants shared between any two independent samples". As Cas9 off-targets are largely sequence-dependent and recurrent, meaning that off-targets are expected to present in more than one individual, what were the rationales and justifications for this secondary filter?

RESPONSE: This secondary filter is necessary to identify off-targets not due to Cas9 activity. Our sample group consists of untreated controls (no Cas9 and/or guide) and treated (Cas9 plus guides); however, none of the treated samples are replicates (i.e., none use the same guide sequence(s)). As each sample is independent, we do not expect a recurrent mutation to be due to Cas9 activity.

"While we did computationally detect unintentional Cas9-mediated off-target activity in 36% (18/50) of our lines, only 9/28 (32%) off-target events, associated with 16% (8/50) lines, were confirmed." Please clarify what "computationally detect" mean exactly, and what was the confirmation method.

RESPONSE: We have added further clarification to the text (pg. 6). Computational detection by our pipeline is based upon overlap between a variant call from WGS data and predicted off-target site allowing up to 6 mismatches. Putative off-target variants associated with small indels and some SVs, where the repetitive nature of the genomic sequence did not preclude appropriate primer design, were subjected to Sanger sequencing in founder DNA. In the revised manuscript, we have also included CIRCLE-seq analysis for 6 guides.

There is a typo "CRISRPR".

RESPONSE: Fixed

Once again, thank you for your thoughtful consideration of our manuscript.

Best regards,

Lauryl M. J. Nutter
Director, Research and Technology Development
The Centre for Phenogenomics

Technical Innovation Investigator
The Hospital for Sick Children

t: 416.813.7654 x309565

e: lauryl.nutter@sickkids.ca

w: phenogenomics.ca

REVIEWERS' COMMENTS:

Reviewer #1 (Remarks to the Author):

The authors have addressed my concerns appropriately in the revised manuscript.

Reviewer #2 (Remarks to the Author):

Rather than repeat the comments from my first review, I will only say that the authors have adequately addressed my concerns in their revision.

The Centre for Phenogenomics
The Hospital for Sick Children
25 Orde Street
Toronto, Ontario M5T 3H7

1 May 2023

RE: Manuscript COMMSBIO-21-2206A

We were happy to see that we had addressed all of the Reviewers' comments and that no additional changes were requested.

Best regards,

Lauryl M. J. Nutter
Director, Research and Technology Development
The Centre for Phenogenomics

Technical Innovation Investigator
The Hospital for Sick Children

t: 416.813.7654 x309565
e: lauryl.nutter@sickkids.ca
w: phenogenomics.ca